# Patient-reported experiences and views on the Cytosponge test: a mixed-methods analysis from the BEST3 trial

Roberta Maroni ,[1] Jessica Barnes,[1] Judith Offman,[2] Fiona Scheibl,[3] Samuel G Smith,[4] Irene Debiram-Beecham,[5] Jo Waller ,[2] Peter Sasieni ,[1,2] Rebecca C Fitzgerald,[5,6] Greg Rubin ,[7] BEST3 Consortium,[8] Fiona M Walter [9,10]

RM, JB and JO contributed equally.

For numbered affiliations see end of article.

**Correspondence to**
Dr Judith Offman;
judith.offman@kcl.ac.uk

## ABSTRACT

**Objectives** The BEST3 trial demonstrated the efficacy and safety of the Cytosponge-trefoil factor 3, a cell collection device coupled with the biomarker trefoil factor 3, as a tool for detecting Barrett's oesophagus, a precursor of oesophageal adenocarcinoma (OAC), in primary care. In this nested study, our aim was to understand patient experiences.

**Design** Mixed-methods using questionnaires (including Inventory to Assess Patient Satisfaction, Spielberger State-Trait Anxiety Inventory-6 and two-item perceived risk) and interviews.

**Outcome measures** Participant satisfaction, anxiety and perceived risk of developing OAC.

**Setting** General practices in England.

**Participants** Patients with acid reflux enrolled in the intervention arm of the BEST3 trial and attending the Cytosponge appointment (N=1750).

**Results** 1488 patients successfully swallowing the Cytosponge completed the follow-up questionnaires, while 30 were interviewed, including some with an unsuccessful swallow.

Overall, participants were satisfied with the Cytosponge test. Several items showed positive ratings, in particular convenience and accessibility, staff's interpersonal skills and perceived technical competence. The most discomfort was reported during the Cytosponge removal, with more than 60% of participants experiencing gagging. Nevertheless, about 80% were willing to have the procedure again or to recommend it to friends; this was true even for participants experiencing discomfort, as confirmed in the interviews.

Median anxiety scores were below the predefined level of clinically significant anxiety and slightly decreased between baseline and follow-up (p < 0.001). Interviews revealed concerns around the ability to swallow, participating in a clinical trial, and waiting for test results. The perceived risk of OAC increased following the Cytosponge appointment (p<0.001). Moreover, interviews suggested that some participants had trouble conceptualising risk and did not understand the relationships between test results, gastro-oesophageal reflux and risk of Barrett's oesophagus and OAC.

**Conclusions** When delivered during a trial in primary care, the Cytosponge is well accepted and causes little anxiety.

## Strengths and limitations of this study

► Our study is the first to explore patients' experiences of, and satisfaction with, the Cytosponge test in the primary care setting, gaining in-depth understanding by using questionnaire and interview data in a mixed-methods approach.

► Thirty participants, purposively sampled to reflect a range of characteristics (gender, age group, geographical region, Cytosponge result), underwent semistructured interviews, whose analyses were underpinned by a robust approach, including a conceptual framework.

► A small proportion (10%) of patients undergoing the Cytosponge test did not complete a follow-up questionnaire.

► The Inventory to Assess Patient Satisfaction in the follow-up questionnaire was adapted from flexible sigmoidoscopy and validated on a small number of individuals.

**Trial registration number** ISRCTN68382401.

## INTRODUCTION

In the UK, oesophageal adenocarcinoma (OAC) is the seventh most common cause of cancer death. It has a bleak prognosis, with a 5-year net survival of just 17%.[1] Most cases of OAC are preceded by Barrett's oesophagus (BO), which provides an opportunity for early detection.[2 3] Besides age, sex (male), obesity, ethnicity (Caucasian) and family history,[4 5] the most important risk factor for BO is gastro-oesophageal reflux disease (GORD). Currently, only around 20% of patients with BO are diagnosed[6] since endoscopy is not feasible for all patients with GORD, and not all patients with GORD experience heartburn symptoms and so they may not come to medical attention. Overall, GORD burdens about 20% of the adult population[7] and is usually managed effectively with

acid-suppressant medications and endoscopy referrals, which are suggested by the National Institute for Health and Clinical Excellence (NICE) only if the symptoms are not controlled.[8] However, endoscopies are invasive, expensive[9] and entail some risks.[10] Given that pressures on endoscopy capacity in secondary care in the UK have been exacerbated by the recent COVID-19 pandemic,[11] novel technologies to help detect BO are now more critical than ever.

The Cytosponge is a cell collection device, which, coupled with the biomarker trefoil factor 3 (TFF3), can be used to identify BO. The device consists of a sponge tied to a string and compressed into a gelatine capsule, which is swallowed by the patient and retrieved by pulling on the string once the capsule has dissolved. The cell sample is then processed in a laboratory for immunohistochemical staining with TFF3. The Cytosponge-TFF3 test has been evaluated among more than 2000 patients in two clinical settings,[12 13] proving its safety, cost-effectiveness and accuracy as a potential test for BO.[14–17] The large (N>13 000), pragmatic, randomised, controlled BEST3 trial was recently conducted in primary care in England[18] and demonstrated that offer of the Cytosponge test to individuals on medication for recurrent reflux symptoms identified ten times more cases of BO than usual care. In this trial, fewer than 10% of participants successfully swallowing the Cytosponge reported any side effects, and those were mainly mild (eg, sore throat).

Successful implementation of a new diagnostic device requires not only evidence on diagnostic accuracy, safety, effectiveness and cost-effectiveness, but also an understanding of patient experience and satisfaction, including the identification of possible barriers to uptake. During the BEST3 trial, patients were invited to receive the Cytosponge by postal letter. Invitation uptake was 24% (1654/6834) and median overall acceptability on an 11-point visual analogue scale from 'completely unacceptable' to 'completely acceptable' was 9 (IQR 8–10).[18] Previous BEST studies[12 13 16 17] reported results on acceptability and one of the studies[17] on anxiety scores. However, this nested mixed-methods study as part of the BEST3 trial investigated patients' experiences of the Cytosponge in primary care, any anxiety caused by the test, and perceived risk of OAC more extensively by means of questionnaires and individual interviews.

## METHODS
### Study design
The design of the BEST3 trial is described in more detail in online supplemental materials and elsewhere.[18 19] It enrolled participants aged 50 or over with GORD symptoms, identified via their general practice prescribing records. For this nested study, only participants in the intervention arm of the trial attending a Cytosponge appointment were included (N=1750). Participants with an 'inadequate' test result (ie, low-confidence negative TFF3, equivocal or processing/technical failure)

were invited to a repeat appointment when possible. All patients with a positive TFF3 result were referred for an endoscopy to establish a diagnosis.

### Patient and public involvement
Patient and public involvement representatives were involved in all stages of the BEST3 trial, including two as members of the BEST3 trial steering committee[19]; they reviewed the protocol and the interview topics, and contributed to the early analysis of the patient interviews. The adapted Inventory to Assess Patient Satisfaction (IAPS) was piloted on eight individuals who had previously had the Cytosponge procedure to check for comprehension.

### Data collection
#### Quantitative data
At the Cytosponge appointment, in each participant's general practice, a nurse collected demographic, anthropometric and clinical data, including the GORD Impact Scale.[20] This is a nine-item assessment of GORD symptoms experienced in the week before the appointment, which was duplicated to also address any symptoms from before patients started taking acid-suppressant medications.

Immediately before having the test, participants were asked to complete the baseline questionnaire, with questions on:
► Education level, smoking/alcohol history, family history of heartburn/BO/cancer.
► A shorter six-item form of the Spielberger State-Trait Anxiety Inventory (STAI-6).[21]
► Perceived risk of OAC, using two items widely applied for other cancer risk assessments: perceived risk compared with a person of the same age (comparative risk) and per cent absolute risk of developing OAC in their lifetime.[22]

A week to 14 days later, participants who successfully swallowed the Cytosponge (1654/1750, 95%) were invited to fill in a follow-up questionnaire. A reminder to complete the questionnaire was sent after 2 weeks, and some received a further reminder. The follow-up questionnaire consisted of:
► The IAPS, with 22 items addressing both positive and negative aspects of the experience, adapted from a study on flexible screening sigmoidoscopy.[23]
► STAI-6.
► Perceived risk of OAC (two items).

#### Qualitative data
Participants were purposively sampled to reflect a range of characteristics: gender, age group (50–59, 60–69, 70–79, 80+), geographical region in England (East, North-East, West) and Cytosponge result at first appointment (positive, negative, low confidence/equivocal, unsuccessful swallow) (see table 1 and online supplemental table 1). Semi-structured interviews were conducted face to face at home or in clinics in the only presence of a female qualitative researcher (FS), with the aim of interviewing

**Table 1** Patient characteristics and GORD impact scale for the three subgroups of participation: attended the Cytosponge appointment and completed the baseline questionnaire ('attenders'); completed the follow-up questionnaire ('follow-up responders') or Interviewed ('interviewees')

| | Completed baseline questionnaire ('attenders') (N=1750) | | Completed follow-up questionnaire ('follow-up responders') (N=1488) | | Interviewed ('interviewees') (N=30) | | P values for χ² test between 'follow-up responders' (N=1488) and 'attenders' who are not 'follow-up responders' (N=262) |
|---|---|---|---|---|---|---|---|
| | N | % | N | % | N | % | |
| **Sex** | | | | | | | |
| Female | 919 | 53 | 782 | 52 | 15 | 50 | 0.985 |
| Male | 830 | 47 | 706 | 47 | 15 | 50 | |
| Missing | 1 | <1 | 0 | 0 | 0 | 0 | |
| **Age group** | | | | | | | |
| 50–59 | 345 | 20 | 285 | 19 | 4 | 13 | 0.028* |
| 60–69 | 596 | 34 | 497 | 33 | 11 | 35 | |
| 70–79 | 647 | 37 | 572 | 38 | 9 | 29 | |
| 80–99 | 161 | 9 | 134 | 9 | 6 | 19 | |
| Missing | 1 | <1 | 0 | 0 | 1 | 3 | |
| **Cytosponge-TFF3 outcome (after repeat test)** | | | | | | | |
| TFF3 negative | 1252 | 72 | 1126 | 76 | 14 | 47 | 0.246† |
| TFF3 positive | 231 | 13 | 213 | 14 | 10 | 33 | |
| Inadequate (equivocal/low-confidence negative/technical or processing failure) | 171 | 10 | 149 | 10 | 2 | 7 | |
| Unsuccessful swallow | 96 | 5 | 0 | 0 | 4 | 13 | |
| **Underwent repeat Cytosponge test** | | | | | | | |
| No | 1560 | 89 | 1322 | 89 | 25 | 83 | 0.338 |
| Yes | 190 | 11 | 166 | 11 | 5 | 17 | |
| **Education level** | | | | | | | |
| School up to 15–16 years of age | 712 | 41 | 605 | 41 | 16 | 53 | 0.104 |
| College or vocational school | 537 | 31 | 455 | 31 | 8 | 27 | |
| Professional training beyond college, university graduate or postgraduate degree | 480 | 27 | 414 | 28 | 4 | 13 | |
| Other or prefer not to say | 21 | 1 | 14 | 1 | 2 | 7 | |
| **Waist-hip ratio** | | | | | | | |
| <0.90 | 685 | 39 | 601 | 40 | 11 | 37 | 0.010* |
| 0.90<0.99 | 686 | 39 | 562 | 38 | 10 | 33 | |
| 0.99+ | 378 | 22 | 324 | 22 | 9 | 30 | |
| Missing | 1 | <1 | 1 | <1 | 0 | 0 | |
| **Comorbidities** | | | | | | | |
| No | 228 | 13 | 182 | 12 | 6 | 20 | 0.018* |
| Yes | 1522 | 87 | 1306 | 88 | 24 | 80 | |
| **Medication duration** | | | | | | | |
| Less than 5 years | 518 | 30 | 431 | 29 | 7 | 23 | 0.166 |
| More than 5 years | 1232 | 70 | 1057 | 71 | 23 | 77 | |
| **Diagnoses** | | | | | | | |
| No Barrett's oesophagus | 1618 | 92 | 1367 | 92 | 26 | 87 | 0.118 |
| Barrett's oesophagus—without dysplasia | 117 | 7 | 106 | 7 | 4 | 13 | |
| Barrett's oesophagus—with dysplasia | 11 | 1 | 11 | 1 | 0 | 0 | |
| Oesophageal adenocarcinoma (stage 1) | 4 | <1 | 4 | <1 | 0 | 0 | |

Continued

**Table 1** Continued

| | Completed baseline questionnaire ('attenders') (N=1750) | | Completed follow-up questionnaire ('follow-up responders') (N=1488) | | Interviewed ('interviewees') (N=30) | | P values for χ² test between 'follow-up responders' (N=1488) and 'attenders' who are not 'follow-up responders' (N=262) |
|---|---|---|---|---|---|---|---|
| | **N** | **%** | **N** | **%** | **N** | **%** | |
| | | | | | | | P values for t-test between 'follow-up responders' (N=1488) and 'attenders' who are not 'follow-up responders' (N=262) |
| GORD Impact Scale—before taking acid-suppressant medications | | | | | | | |
| Mean (SD) | 1.9 (0.5) | | 1.9 (0.5) | | 1.9 (0.5) | | 0.319 |
| No. missing | 2 | | 1 | | 0 | | |
| GORD Impact Scale—In the last week | | | | | | | |
| Mean (SD) | 1.3 (0.4) | | 1.3 (0.4) | | 1.3 (0.5) | | 0.451 |
| No missing | 0 | | 0 | | 0 | | |

*P<0.05.
†Comparison excluding participants producing an unsuccessful swallow as they were not invited to fill in a follow-up questionnaire.
GORD, gastro-oesophageal reflux disease ; TFF3, trefoil factor 3.

30 participants within 6 weeks of their Cytosponge test (to reduce issues with recall). Interviews lasted 23 min on average (range 13–50 min), were audio-recorded and transcribed verbatim for analysis. More details on qualitative data collection are in online supplemental materials.

## Analysis
### Quantitative analysis
Questionnaire scoring is described in online supplemental materials. Patient characteristics and GORD Impact Scale responses were analysed according to three subgroups of participation: attended the Cytosponge test appointment and completed the baseline questionnaire ('attenders'); 'attenders' who completed the follow-up questionnaire ('follow-up responders'); 'attenders' who undertook an interview ('interviewees') (see online supplemental figure 1).

STAI-6 and perceived risk of OAC are presented only for the subgroup of participants completing at least one of those items in both baseline and follow-up questionnaires. STAI-6 scores between the two time points were compared using Wilcoxon matched-pairs signed-rank test, while differences in risk perceptions were analysed by McNemar's test, which included only patients with scores different than the neutral (eg, 'neither higher or lower') or middle-ranking category (from a list of ordered options). A STAI-6 score over 40 was predefined as a threshold for clinically significant anxiety.[12 24]

Statistical significance was based on a two-tailed test with size of 5%. Analyses were performed using Stata V.15.[25]

### Qualitative analysis
We undertook a thematic analysis, having organised and managed data according to the Framework approach.[26] For more details, see online supplemental materials. Briefly, this involved identifying an initial, broad set of labels inductively and deductively that would be used to categorise and sort the data to enable the subsequent thematic analysis. Inductively created labels were based on emergent concepts identified in the data. Deductively created labels were based on the IAPS,[23] which allowed us to more directly relate participant experience across qualitative and quantitative datasets. Use of the theoretical framework of acceptability[27] constructs allowed us to examine additional dimensions of patient experience associated with acceptability that were not captured by the IAPS. We then conducted the thematic analysis, aiming to achieve both description and explanation with the dataset. Data within each column of the Framework matrix was explored and further organised into more abstract themes, using drawings.net open-source software to allow visual representation and coding of the data, therefore facilitating the identification patterns and linkages between different types of participant experience and/or demographic characteristics. Participants did not provide feedback on the findings.

## RESULTS
### Demographics and baseline characteristics
A trial flowchart from the intervention arm of the BEST3 trial is shown in online supplemental figure 1. There were 1750 participants who completed the baseline questionnaire at Cytosponge appointment ('attenders'), with a minimum completion rate of 80% (12/15), considering only questions applicable to all participants. The follow-up questionnaire was completed by 1488 participants (90% of 1654 successful swallows) ('follow-up responders'), with a minimum completion rate of 23% (7/31) at a median of 10 days (IQR 7–14 days) after undergoing the Cytosponge test. A total of 159 participants (11% of 1488) completed the follow-up questionnaire after being mailed the letter with their Cytosponge test results, 5

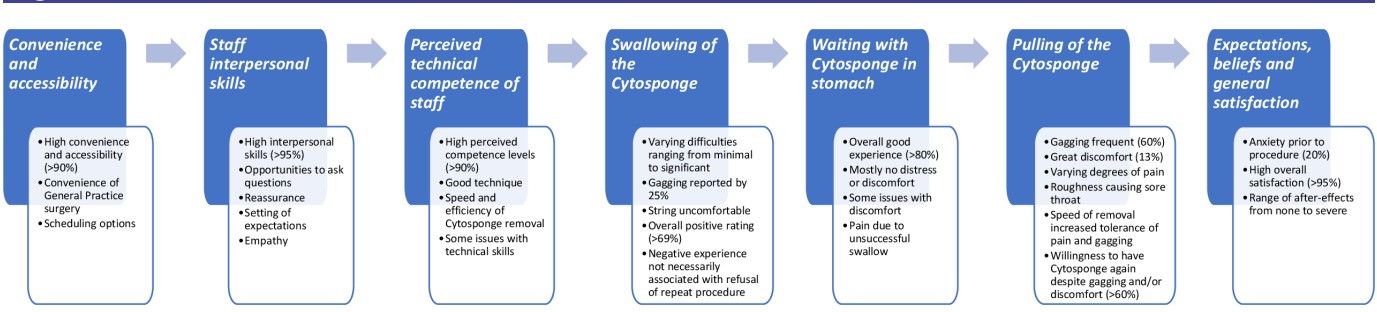

**Figure 1** Summary of findings from questionnaires and patient interviews according to the themes of the Inventory to Assess Patient Satisfaction.

(0.3%) after attending a repeat Cytosponge test and one after receiving their repeat Cytosponge test result.

Out of the 1750 'attenders', 75 (4%) were invited for an interview; 30 interviews were completed ('interviewees') at a median of 59 days (IQR 48–78 days) after the Cytosponge test. At the time of interview, all participants who successfully swallowed the Cytosponge had received their first test result, and one was still waiting for the result from their repeat test, while another had declined to have a repeat test. Among participants who had received a positive test result, some were awaiting confirmatory endoscopy, while others had already had theirs.

Table 1 shows patient and clinical characteristics for the three subgroups of participants. Those completing the follow-up questionnaire differed from the non-completers (N=262) by age group, waist-hip ratio categories and comorbidity status.

### Patient-reported satisfaction and experiences of the Cytosponge test

Participants were generally satisfied with their experiences of the processes undertaken before, during and immediately after the Cytosponge test (figure 1 and table 2). Several items of the IAPS were rated positively by most participants (table 3). The Cronbach's α, measuring the IAPS reliability, was 0.83 overall, and it ranged between 0.81 and 0.83 (improving the overall coefficient in three instances) when excluding each of the 22 items at a time.

### Convenience and accessibility

The majority of participants (92%–94%) rated the study sites' convenience and accessibility positively. In the interviews, some commented that it was practical to go to their own general practice and that this was preferable to going to a hospital appointment. Participants also appreciated the scheduling, as they were able to select from a range of appointment dates and times that suited them.

### Staff interpersonal skills

Staff interpersonal skills were rated positively by 96%–98% of participants, and uniformly described across the interviews in very good terms. Participants felt that they had adequate opportunities to ask questions, which the nurses were able to answer well providing important reassurance. The interpersonal manner of staff was consistently described in highly positive ways by

participants—staff were 'calm', 'in control', 'friendly', 'helpful', 'supportive', 'professional' and created an experience that was 'straightforward and bordering on enjoyable'. When participants failed to swallow the Cytosponge, staff were empathetic and reassuring.

### Perceived technical competence of staff

The majority of participants (93%–96%) agreed that the staff was competent. In the interviews, participants focused on the speed and efficiency of the Cytosponge removal—for example, that the staff members had good technique, and they went quickly enough to get removal done efficiently but slowly enough to gather cells. Some issues related to technical skills were noted: one participant described how their procedure was performed by a practice nurse, and the Cytosponge seemed to get stuck partway through removal and caused pain. In this case, the practice nurse needed to consult a research nurse who advised how to resolve the issue.

### Swallowing the Cytosponge capsule

The majority of participants (69%–87%) rated swallowing the Cytosponge positively. Among the lowest rated measure of satisfaction was 'I had to gag when I swallowed the Cytosponge capsule' (N=373, 25% agreed).

When interview participants described the procedure as straightforward, they recounted the swallowing aspect as routine and nothing unexpected.

> I think that's just normal as taking a tablet, the only difference is it's got string on it. [age 60-69, negative result]

When participants described the swallowing as involving minimal difficulty, they commented on characteristics such as the string being uncomfortable, or that it was difficult to drink enough water to get the string and capsule down. However, these difficulties were mainly perceived as nothing to worry about.

Interviewees who reported significant difficulty swallowing, such as gagging, retching or heaving, underlined issues with being unable to place the capsule and string far enough at the back of their throat without causing themselves to gag; in some cases, this could be rectified by the nurse placing the string and capsule instead of the participant. Participants who failed to swallow reported

**Table 2** Findings from questionnaires and patient interviews: example interview quotes illustrating the practical elements of the Cytosponge procedure

| Aspect of cytosponge procedure | Example interview quotes |
|---|---|
| Convenience and accessibility | *Convenient alternative to a procedure in secondary care:*<br>'(…)from what (the nurse) was saying to me is that (the Cytosponge procedure) takes away that waiting for a hospital appointment, that you can have it done in the (GP) surgery, and if it was me again and, I don't know, something was not quite right, I wouldn't hesitate at coming down and having that done. Not at all, not at all.'<br>*(age 60–69, inadequate test result at first appointment)* |
| Staff interpersonal skills | *Positive interpersonal skills:*<br>'The (nurse) who actually did it was really lovely. She really was. She was very calm, very in control and we chatted about different things and she was about to get married and all this sort of thing and it was, we learnt a bit about each other, which was absolutely fine(…)(The procedure is) done very nicely, lovely people, nothing to worry about, go and get it done.'<br>*(age 70–79, negative test result)*<br>*Procedure explained clearly:*<br>'I mean if I didn't understand then I asked to explain it. I think (the staff) were very helpful and very nice, the way they put things over. I mean there was the two of them here and what one didn't answer, the other one did. No, I think they were very helpful and very kind.'<br>*(age 80 and over, failed swallow)* |
| Perceived technical competence of staff | *Staff were skilled at removing sponge:*<br>'No, it was fine, it was just that and she did really well, she (removed the Cytosponge) as quick as she could be, obviously she had to go slower to get what she needed.'<br>*(age 50–59, negative test result)*<br>*Patients and inexperienced staff may need more guidance:*<br>'…the first part of the extraction (of the Cytosponge) was fairly non-event(ful) but then again it did get stuck a bit in my throat(…)And the (practice) nurse had to ask the (research) nurse(…)she just said pull harder. So she pulled harder and it popped out.(…)So I don't know if positioning the throat in a different way or me being told to hold the throat in a particular angle may have helped but, I mean, I know that sword swallowers, they hold their throat quite straight(…)But there was no advice as to how to hold your head or position your throat and I thought that might have been useful(…)Well to hold the head in a particular position and relax may have helped, I don't know, it may have got stuck whatever.'<br>*(age 70–79, positive test result)* |
| Swallowing of the Cytosponge | *Difficulties due to string and retching:*<br>'The first time, when I swallowed it, the string seemed to flick around in the back of my throat and it didn't go down properly, so I was trying to add a bit more water and that, but I couldn't(…)I was just retching all the time and I couldn't even get(…)the water in my mouth because I just kept retching all the time(…)And then the second time, it went straight down, straight down. It was marvellous, it went straight down and I thought, oh, I've cracked it, so I just kept sipping, and then all of a sudden I think a bit of the string… Like I felt down at the side, and I just went uh, and it just came straight out, just all came straight out altogether.(…)I think it's the water I drank, it was still lying on my stomach and just brought it straight back up.'<br>*(age 50–59, failed swallow)*<br>*Swallowing was easy:*<br>'That swallowing the capsule was simple, there was no… it was easy, it was just a matter of a few mouthfuls of water and that was it.'<br>*(age 50–59, positive test result)* |
| Waiting with Cytosponge in stomach | *Waiting was acceptable, especially when there were distractions:*<br>'But it wasn't horrendous and for the time that I was there and, you know, and by the time I'd sort of swallowed (the Cytosponge) and answered a few questions, had a little chat and drank some water, it was time for it to come up.'<br>*(age 60–69, negative test result)*<br>*Mild discomfort:*<br>'You're aware of the string being in the throat(…)It was slightly uncomfortable(…)It was making you want to (cough) (…)(but) There was no problem with it.'<br>*(age 60–60, negative test result)* |
| Pulling of the Cytosponge | *Experience of pain:*<br>'It was painful. It was worse than I was expecting(…)the nurse explained it to me afterwards, because afterwards I said to her, I said wow, that was more painful than I was expecting, and she explained that where your muscles will work to push things downwards, obviously, she said, when you're pulling the sponge up you're going completely against everything that it's doing and, do you know, I couldn't even describe what sort of pain it was, but it was literally… well it felt like a sponge literally was pulling out, you know, but(…)I have to say, it only lasted a few seconds, and once it was out I suppose I had a tickly throat for the rest of the day. Not hurting, just a bit scratchy, tickly, certainly no painkillers, nothing like that. It was just those few seconds of it actually coming out wasn't pleasant, no.(…)I did come back again (for second appointment following inadequate test result).'<br>*(age 60–69, inadequate test result)*<br>*Discomfort from gagging/coughing:*<br>'It was all over in a matter of seconds, but it was when it hit the back of my throat, I did gag, and I started to cough or I had a coughing fit after it was out, I was red hot, you know, I think it was just with gagging, yeah, but it was fine, it was just something that lasted a matter of two seconds.'<br>*(age 50–59, positive test result)*<br>*No discomfort:*<br>'It was over and done within a matter of…(…)Woosh, gone.(…)Finished, I didn't even feel it coming out.'<br>*(age 60–69, negative test result)* |

**Table 3** Number and proportion of participants (N=1488) by ratings for the 22 questions of the inventory to assess patient satisfaction

| | Disagree | | Neither | | Agree | | Missing | |
|---|---|---|---|---|---|---|---|---|
| | N | % | N | % | N | % | N | % |
| **Convenience and accessibility** | | | | | | | | |
| I did not feel that I had to wait too long.* | 42 | 3 | 24 | 2 | 1395 | 94 | 28 | 2 |
| The test is in a place that is easy for me to get to. | 90 | 6 | 4 | <1 | 1389 | 93 | 5 | <1 |
| I did not find it hard to find a convenient time to come to the test.* | 71 | 5 | 15 | 1 | 1368 | 92 | 34 | 2 |
| **Staff interpersonal skills** | | | | | | | | |
| I felt free to ask the staff questions I wanted to ask. | 23 | 2 | 1 | <1 | 1456 | 98 | 8 | 1 |
| The staff did not seem to hurry me through too quickly.* | 9 | 1 | 2 | <1 | 1454 | 98 | 23 | 2 |
| The staff did not use words that were hard to understand.* | 22 | 1 | 10 | 1 | 1425 | 96 | 31 | 2 |
| **Perceived technical competence** | | | | | | | | |
| The nurse or member of staff was not too rough when performing the Cytosponge test.* | 20 | 1 | 14 | 1 | 1422 | 96 | 32 | 2 |
| I feel confident that the Cytosponge test was performed properly. | 86 | 6 | 10 | 1 | 1384 | 93 | 8 | 1 |
| **Swallowing of the capsule** | | | | | | | | |
| I did not have to gag when I swallowed the Cytosponge capsule.* | 373 | 25 | 53 | 4 | 1020 | 69 | 42 | 3 |
| Swallowing the Cytosponge capsule was more comfortable than I expected. | 221 | 15 | 169 | 11 | 1073 | 72 | 25 | 2 |
| Swallowing the Cytosponge capsule did not cause me great discomfort.* | 82 | 6 | 60 | 4 | 1300 | 87 | 46 | 3 |
| **Waiting with the capsule in stomach** | | | | | | | | |
| I did not have to gag while I waited with the Cytosponge capsule in my stomach.* | 146 | 10 | 36 | 2 | 1264 | 85 | 42 | 3 |
| Waiting with the Cytosponge capsule in my stomach was more comfortable than I expected. | 123 | 8 | 133 | 9 | 1207 | 81 | 25 | 2 |
| Waiting with the Cytosponge capsule in my stomach did not cause me great discomfort.* | 39 | 3 | 36 | 2 | 1365 | 92 | 48 | 3 |
| **Pulling of the Cytosponge** | | | | | | | | |
| I did not have to gag when the Cytosponge was pulled up.* | 889 | 60 | 68 | 5 | 491 | 33 | 40 | 3 |
| Pulling up of the Cytosponge was more comfortable than I expected. | 354 | 24 | 234 | 16 | 866 | 58 | 34 | 2 |
| Pulling up of the Cytosponge did not cause me great discomfort.* | 193 | 13 | 108 | 7 | 1134 | 76 | 53 | 4 |
| **Expectations and beliefs** | | | | | | | | |
| I was not very anxious about having the Cytosponge test.* | 296 | 20 | 132 | 9 | 1029 | 69 | 31 | 2 |
| Undergoing the Cytosponge test will benefit my health. | 27 | 2 | 281 | 19 | 1153 | 77 | 27 | 2 |
| **General satisfaction** | | | | | | | | |
| I was very satisfied with the care I received. | 16 | 1 | 2 | <1 | 1450 | 97 | 20 | 1 |
| I would recommend the Cytosponge test to my friends. | 38 | 3 | 184 | 12 | 1236 | 83 | 30 | 2 |
| I would be willing to have another test if necessary. (As part of the Trial, you may still be contacted for a repeat Cytosponge test.) | 48 | 3 | 229 | 15 | 1185 | 80 | 26 | 2 |

*Items referring to negative aspects of patient experience were rephrased for this table using negative constructs to facilitate comparison between items.

struggling with getting the string down as it unwound, and gagging too much to be able to drink water to wash the string and capsule down the oesophagus. One participant reported that they had not realised that they would be required to swallow the string in a bundle, and if they had known this, they may have declined to participate. During the interviews, responses were varied among the four participants who failed to swallow: some would not

do the procedure again but would still recommend it to friends, while others would still try the procedure again in the future.

> Well swallowing the capsule was all right. The string attached to it was a bit difficult, it felt a bit like a cat trying to swallow a mouse, you know, can't get the tail in the mouth. […] It went down all right… it was just an odd feeling with the string coming up. [age 60-69, negative result]

### Waiting with the Cytosponge in stomach

Overall, 85%–92% participants rated the experience of waiting for the capsule to dissolve in their stomach positively. During the interviews, some reported not being able to feel anything untoward at all, nor did they experience any distress. Others reported minor issues, such as being aware of the string, tickling or gagging when trying to talk, but these experiences were not considered concerning.

> The only strange sensation was… after I'd swallowed the pill, it was having a tiny piece of cotton or whatever it was hanging out, but the way [the nurses] talked, it took my mind off it anyway. [age 60-69, negative result]

Some participants discussed more distressing experiences: one experienced 'pains in my stomach', significant enough for them to ask the nurse to remove the Cytosponge prematurely. This procedure resulted in a low confidence/equivocal result, and the participant attended a repeat appointment, where they were able to successfully swallow the Cytosponge and it was 'less uncomfortable' while the capsule was dissolving. The participant suggested that this may be because they drank more water the second time, causing the Cytosponge to successfully reach the stomach.

### Retrieving the Cytosponge

Among the lowest rated measures of satisfaction were: 'I had to gag when the Cytosponge was pulled up' (N=889, 60% agreed) and 'Pulling up of the Cytosponge was more comfortable than I expected' (N=354, 24% disagreed).

During the interviews, participants gave more detailed descriptions of this part of the Cytosponge test, with some reporting a number of types of discomfort during removal (figure 1 and online supplemental box 1). Not all of them were particularly serious or concerning. Despite these experiences of discomfort, participants often expressed a willingness to have the Cytosponge test again and to recommend it to others, as confirmed by responses to the IAPS questionnaire (61% and 65% of participants with low average satisfaction scores for items about pulling the Cytosponge, respectively). One interviewee was unwilling to have the procedure again due to the perceived possibility of the string breaking and an endoscopy being necessary to retrieve it.

### Expectations, beliefs and general satisfaction

A fifth of the participants (20%) agreed with the item 'I was very anxious about having the Cytosponge test', while 97% reported being very satisfied with the care received.

During the interviews, participants discussed a range of after-effects (including none). Some explained that they felt fine after their appointment, and sometimes forgot completely about it until they received their results letter. Some participants described experiencing a sore, scratchy or tickly throat that resolved relatively quickly. Some reported experiencing unexpected reflux following their test.

Experiences linked to the understanding of the test results, discussed during the patient interviews, are presented in online supplemental table 2. On receiving their Cytosponge-TFF3 test results, participants reacted in ways that were influenced by their expectations, which varied due to a number of interplaying factors. These included: their understanding of the purpose of the test; previous relevant experiences that had improved their literacy of such test results; and their conceptualisation of the causes of cancer in general.

Some participants receiving a positive test result reacted with shock as the result went against their expectations, which were based on their understanding of the causes of cancer in general: their explanations of their reaction to their test result revealed an assumption that a positive result should only be expected by people who have a particular lifestyle or risk factors (such as a history of drinking or smoking), or if BO or oesophageal cancer runs in the family. Other participants who reacted with shock to a positive test result described that the trial's reference to cancer was heightened in their mind, so receiving a positive result was experienced to some degree as being like receiving a cancer diagnosis.

There was an issue with the language that was used to report test results, which caused confusion and concern. For positive test results, the issue was around use of the term 'positive' as some participants initially interpreted this in the lay sense as meaning 'good'. Alternative terminology such as 'needs further investigation' was recommended. Another issue with phrasing was around the reassurances that a positive result was 'nothing to be unduly concerned about'. Participants explained that, paradoxically, this made them more concerned, and they felt that they were not given adequate information about what a positive result meant to enable them to understand why exactly they should not be concerned.

### Patient-reported anxiety before and after the Cytosponge test

Participants who completed both pre-test and post-test measures (N=1418) had a median STAI-6 score of 33 (IQR 23–40; possible range: 20–80) at baseline and 27 (IQR 20–37) at follow-up. As a comparison, the median score for participants not filling in the follow-up questionnaire was 33 at baseline (IQR 23–43). A score of over 40 was predefined as meeting a clinical threshold of anxiety: 334 (24%) and 166 (12%) reported such scores at baseline

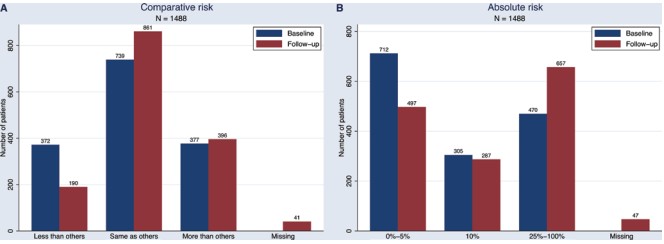

**Figure 2** Ratings for perceived risk of oesophageal cancer at the Cytosponge appointment (baseline) and 7–14 days follow-up for participants completing both baseline and follow-up questionnaires (N=1488). (A) Risk of oesophageal cancer compared with someone of the same age (comparative risk). (B) Per cent absolute risk of oesophageal cancer. Possible answers to the multiple-choice question on absolute percent risk of oesophageal cancer were: 0%, 5%, 10%, 25%, 50%, 75% and 100%. Participants with missing answers on perceived risk of oesophageal cancer at follow-up were still included in the figures as they filled in other parts of the questionnaire.

and follow-up, respectively. There was a statistically significant difference in scores between baseline and follow-up (p<0.001), with a median change between follow-up and baseline of −3 (IQR −10–0). For a breakdown of scores by questionnaire, see online supplemental table 3.

Interviewees offered reflections on how they were feeling the day of the appointment or the night before. Some were worried about being able to complete the test (eg, participants who had problems swallowing) or the test itself. A few were concerned about 'the unknown' and it being 'experimental'. Reflecting on the period after the Cytosponge test, some described how receiving their result alleviated the sense of anxiety and uncertainty that they had been experiencing. Other participants, however, reported not being particularly bothered while waiting for their results.

### Perceived risk of oesophageal cancer

Among participants filling in both questionnaires, just over half (N=739, 50% at baseline; N=861, 58% at follow-up) considered their risk to be equivalent to that of someone of the same age (see figure 2A for a comparison of the ratings between baseline and follow-on questionnaire). Opinions on absolute risk in a lifetime were more varied: while the largest group pre-test (N=712, 48%) thought that their risk was not more than 5%, the largest group post-test (N=657, 44%) expected theirs to be higher than 25% (see figure 2B for a comparison of the ratings pre-test and post-test). There was a statistically significant change (p<0.001) for both items of perceived risk between baseline and follow-up, with 319 (21%) and 389 (26%) participants thinking that their chances of OAC had increased for comparative and absolute risk, respectively (online supplemental tables 4 and 5).

Some interview participants did not demonstrate a good understanding of the relationship between reflux, BO and OAC, which may have led them to different interpretations of questions about personal risk of BO and OAC, and the size of their risk:

> …when I was searching for the probability, the ratio of Barrett's to actual oesophageal cancer, I seemed to be getting different answers. [age 70-79, positive test result]

Some participants found the information about risk in the invitation leaflet difficult to understand but drew attention to the important role that the nurses played in explaining this to them at the start of the appointment.

> Because I'd never heard of Barrett's before,… obviously it was on the leaflets I read that, but when I actually come for the test the nurse that I saw… she explained it all to me and… how that can be a sign that you may get the cancerous cells and things like that. So yeah, it was very interesting. I didn't know that. [age 60-69, inadequate test result]

## DISCUSSION

### Summary of main findings

This mixed-methods study evaluated patients' experiences of, and satisfaction with, the Cytosponge test in primary care as part of the BEST3 trial. Overall, participants were satisfied with their experience of the Cytosponge: they found it very convenient to attend their appointment at their own general practice and rated the staff interpersonal skills and competence very highly. Regarding the Cytosponge procedure itself, 87% of participants did not find swallowing very uncomfortable, while 60% reported gagging during the Cytosponge withdrawal in the questionnaire data; despite that, more than 80% were willing to have the test again or to recommend it to others. In interviews, patients provided more detailed descriptions of their experience, specifically different levels of pain and scratching resulting in a sore throat. Questionnaire data found a slight decrease in anxiety levels between before and after the test, and interviews helped identify patients' underlying motivations for feeling anxious: their ability to swallow, participating in a clinical trial, and waiting for test results. Lastly, we observed a statistically significant change in perceived risk of OAC pre-test and post-test with 21%–26% (depending on the risk type) of participants rating their risk as higher at follow-up. Interview data suggested that information about risk in the invitation leaflet was difficult to interpret for some patients and that nurses played an important role in providing more information on risk at the appointment, despite participants still not having a good understanding of the relationship between reflux, BO and OAC.

### Interpretation

This study has provided a deeper understanding of those aspects of the Cytosponge test that worked well in the trial and would need to be maintained to ensure acceptability during implementation. First, delivering the Cytosponge

near home was perceived as convenient and acceptable. Second, the nurses administering the Cytosponge were rated as supportive, knowledgeable and reassuring. Third, staff technical competence was also rated very highly. Implementation of the Cytosponge test as a routine diagnostic test in primary care will need to ensure balance between convenience and adequate staff training.

Some aspects of the Cytosponge test were rated less well and our interview data provide insights into what could be changed. First, although the majority of participants (95%) were able to successfully swallow the device, swallowing and retrieval of the Cytosponge were less highly rated. Despite experiencing different levels of discomfort, most participants found that the pain was as expected, suggesting that it is important to explain how the Cytosponge is removed, that removal is brief, and that some discomfort may be necessary for the sponge to effectively gather cells. To ensure a good overall experience continues when implementing the Cytosponge in primary care, it will be important to provide high-quality information and manage patient expectations of the physical experience, as was done in BEST3. This was achieved by explaining the procedure using the BEST3 leaflet and a demo Cytosponge as support, and reassuring about the potential risks at the beginning of the appointment. Second, some interviewees reported varying levels of pre-test anxiety linked to concerns about swallowing the device and general fear of the unknown. However, the median STAI-6 scores observed before the procedure and at follow-up were both well under the predefined level of clinically significant anxiety of 40 in the average adult population in a non-clinical setting.[24] In some cases, pre-test anxiety might improve once patients are more familiar with the Cytosponge. While these findings are broadly reassuring, efforts should be made to ensure patients know what to expect and are supported if they feel anxious. Using the same leaflet as in BEST3 and a demo Cytosponge, this should be achievable within the time frame available in standard clinical National Health Service (NHS) practice.

At both time points, the majority of participants rated their risk of OAC as being average for people of their age, showing some evidence of the 'optimistic bias' often observed in measures of comparative risk. At follow-up, a greater proportion of people rated their risk as being above average, which may reflect a greater awareness of the association between reflux, BO and OAC following the procedure. However, the qualitative data point to an inconsistent understanding of the relationships between these three conditions. There are some parallels with the cervical screening context: confusion about the relationship between human papillomavirus (HPV), cervical intraepithelial neoplasia and cervical cancer is common and many women receiving a positive HPV result report adverse psychological outcomes.[28] As Cytosponge testing is rolled out more widely, it will be important to use best practice in risk communication[29] to ensure people understand the meaning of results to minimise misunderstanding and poor psychological outcomes.

## Context of other literature

Previously, acceptability of the Cytosponge test had been assessed using a visual analogue scale ranging from 0 (worst) to 10 (best experience).[12 13 30] A review of five studies assessing the Cytosponge test found a satisfactory overall acceptability, with a median score of 6.[16] In addition, the BEST1 study showed, using the STAI-6, that anxiety levels were low before and after the test with similar scores obtained as in this current study.[12] One qualitative study has investigated the acceptability of Cytosponge, but the participants had not actually taken the test, so their attitudes were hypothetical.[31] It showed that acceptability was high despite initial concerns about swallowing and extracting the capsule.

Even though BEST3 participants experienced different levels of discomfort or pain during the swallowing and removal stages of the procedure, in most cases this would not discourage them from having the test again or recommending it to someone else. This is relevant in the context of implementing the Cytosponge as a routine test. Interestingly, studies investigating barriers to screening attendance found varying degrees of association between pain and reattendance, with 25%–46% of women citing pain of having a mammography as a reason for non-attendance[32]; however, worry about pain was not associated with low intention to reattend cervical screening.[33]

## Strengths and limitations

This study was undertaken within a large pragmatic randomised controlled trial, in which 1750 patients attended the Cytosponge appointment. Key strengths are that the BEST3 trial was set in primary care, where Cytosponge implementation is planned, and that this study used a mixed-methods approach. The findings from the IAPS, STAI-6 and perceived risk questionnaires, completed by nearly 1500 participants, were explored in more depth during interviews with a diverse sample of 30 patients, which included patients with unsuccessful swallows whose experience had otherwise not been captured in the follow-up questionnaire. The qualitative analyses, supported by a conceptual framework, offered detailed insights of participants' experiences and enriched the interpretation of the quantitative findings.

This study had limitations. Some attendees (N=262, 15%) did not return the follow-up questionnaire, and there were some small statistically significant differences in the distribution of patients' characteristics in those completing vs not completing the follow-up questionnaire. However, a simulation including the non-completers and assuming that they had given the worst ratings to their Cytosponge experience in the IAPS and STAI-6 questionnaires showed good overall levels of patient satisfaction (about 80%) and relatively low levels of anxiety (median 30, IQR 20–43, results not shown).

The IAPS, which had been adapted from flexible sigmoidoscopy, was only validated by piloting with a small number of patients, but the Cronbach's alpha of 0.83 indicates appropriate internal reliability of the adaptation of the questionnaire to the Cytosponge test. The predefined threshold of clinical anxiety (over 40) used in our analysis for the STAI-6 was defined in the literature for a non-clinical setting and for the complete STAI questionnaire.[24] The main limitation of the qualitative findings was that, for some, more than 6 weeks elapsed from a participant's Cytosponge procedure and their interview. This may have affected recall, although most participants were able to remember their experiences in substantial detail.

## CONCLUSION

This study, exploring patients' experiences of, and satisfaction with, the Cytosponge test used extensive questionnaire and in-depth interview data. Overall, participants were satisfied with their experiences and we did not observe excess anxiety due to the procedure. Identifying aspects of the procedure which are currently working well or rated less positively will enable specific improvements to communications with patients, for example on how to better communicate test results, that will result in a better experience once the Cytosponge test is implemented in clinical care.

**Author affiliations**
[1]Cancer Research UK and King's College London Cancer Prevention Trials Unit (CPTU), Cancer Prevention Group, School of Cancer & Pharmaceutical Sciences, King's College London, London, UK
[2]Cancer Prevention Group, School of Cancer and Pharmaceutical Sciences, King's College London, London, UK
[3]Norwich Medical School, University of East Anglia, Norwich, UK
[4]Leeds Institute of Health Sciences, University of Leeds, Leeds, UK
[5]MRC Cancer Unit, Hutchison-MRC Research Centre, University of Cambridge, Cambridge, UK
[6]Cambridge University Hospitals NHS Foundation Trust, Cambridge, UK
[7]Population Health Sciences Institute, Newcastle University, Newcastle upon Tyne, UK
[8]BEST3 Trial team NIHR, Clinical Research Networks, UK
[9]The Primary Care Unit, Department of Public Health & Primary Care, University of Cambridge, Cambridge, UK
[10]Wolfson Institute of Population Health, Queen Mary University of London, London, UK

**Acknowledgements** This research is linked to the CanTest Collaborative, which is funded by Cancer Research UK (C8640/A23385), of which FMW is Director and GR is Chair of the Steering Committee. We would also like to acknowledge patient input from Heartburn Cancer UK, and the contributions of our CanTest PPI lead, Mrs Margaret Johnson. We acknowledge the contribution of statistician Irene Kaimi, who had a leading role early in the trial but who tragically died before the study was completed. We thank all of the sites and patients who participated in the BEST3 trial, without whom this research would not have been possible.

**Contributors** RCF was the chief investigator of the BEST3 trial. GR and FMW conceptualised and JO, GR and FMW designed this mixed-methods study as nested within the BEST3 trial. FS conducted the patient interviews. ID-B, RM, JB and FS acquired and analysed the data. RM, JB, JO, SS, JW, PS and FMW interpreted the data. RM, JB, JO and FMW drafted the manuscript. PS provided statistical support.

FMW takes responsibility for the overall content as the guarantor. All authors critically reviewed the manuscript and approved the final version.

**Funding** The laboratory of RCF is funded by a Core Programme Grant from the Medical Research Council (RG84369). The BEST3 study was primarily funded by Cancer Research UK (C14478/A21047). The National Institute for Health Research (NIHR) covered service support costs and National Health Service commissioners funded excess treatment costs, while Medtronic funded Cytosponge devices and TFF3 antibodies. This research was supported by the NIHR Cambridge Biomedical Research Centre (BRC-1215–20014). Cancer Research UK provide funding to the Cancer Prevention Trials Unit (C8162/A25356) and Cancer Prevention Group at King's College London (C8162/A16892). SS is supported by a Yorkshire Cancer Research Fellowship (L389SS). JW is funded by a Cancer Research UK career development fellowship (C7492/A17219).

**Competing interests** RCF is named on patents related to the Cytosponge-trefoil factor 3 test. Covidien GI Solutions (now Medtronic) licensed the Cytosponge from the Medical Research Council, and the device has now received the CE mark and is cleared by the US Food and Drug Administration. RCF is a shareholder in Cyted, a company working on early detection technology. PS reports fees paid to his organisation from GRAIL, outside of the submitted work.

**Patient and public involvement** Patients and/or the public were involved in the design, or conduct, or reporting, or dissemination plans of this research. Refer to the Methods section for further details.

**Patient consent for publication** Not applicable.

**Ethics approval** This study involves human participants and was approved by East of England – Cambridge East Research Ethics Committee.

**Provenance and peer review** Not commissioned; externally peer reviewed.

**Data availability statement** Data are available on reasonable request. The trial protocol, statistical analysis plan and statistical report will be available via the University of Cambridge data repository (https://www.data.cam.ac.uk/repository). Datasets will be available from the authors on request.

**ORCID iDs**
Roberta Maroni http://orcid.org/0000-0001-6420-2881
Jo Waller http://orcid.org/0000-0003-4025-9132
Peter Sasieni http://orcid.org/0000-0003-1509-8744
Greg Rubin http://orcid.org/0000-0002-4967-0297
Fiona M Walter http://orcid.org/0000-0002-7191-6476

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
