## [Reviewer comments · BMJ Open]

ARTICLE DETAILS

TITLE (PROVISIONAL)	Patient-reported experiences and views on the Cytosponge test: a mixed-methods analysis from the BEST3 trial
AUTHORS	Maroni, Roberta; Barnes, Jessica; Offman, Judith; Scheibl, Fiona; Smith, Samuel; Debiram-Beecham, Irene; Waller, Jo; Sasieni, Peter; Fitzgerald, RC; Rubin, Greg; Walter, Fiona

VERSION 1 – REVIEW

REVIEWER	Mukherjee, Swarupananda NSHM Knowledge Campus - Kolkata, Pharmaceutical Technology
REVIEW RETURNED	04-Aug-2021

GENERAL COMMENTS	The stated objective of the work presented in the manuscript was Patient-reported experiences and views on the Cytosponge test: a mixed-methods analysis from the BEST3 trial. . While there was a large amount of data presented in the manuscript, there are SOME MINOR deficiencies and I have several questions and observations; as outlined below: 1. The part of discussion (INTERPRETATION PART) needs to be modified as the logic of this section is not clear and the contents are not closely related to the experimental results.2. It would be better if the author could provide more detailed information at the beginning (RATIONALITY OF THE OBJECTIVES) of the article about the previous approaches.3. Some spelling mistakes can't be overlooked. Few plagiarized words and sentences need to be modified.4. Figure No. 2 require better resolution AND caption need to be modified.5. (a) Risk compared to someone of the same age (comparative risk). (b) Percent absolute risk. from Fig 2 need to be explained clearly in the discussion part of the manuscript.6. Novelty of the work should be discussed in the introduction part. Remaining part is ok and can be considered for publication.
---

REVIEWER	Nicholson, Brian University of Oxford, Nuffield Dept Primary Care Health Sciences
REVIEW RETURNED	30-Nov-2021

GENERAL COMMENTS	Cytosponge offers significant promise to detect Barrett's Oesophagus in community settings remote from traditional invasive endoscopy. This mixed methods analysis examines in detail patients' experience of using Cytosponge, associated anxiety, and perceived cancer nested in the BEST3 trial, the largest trial of Cytosponge in primary care to date. These data are of vital importance as the thought of swallowing a capsule on a
---

	string then dragging a sponge back up the foodpipe is terrifying to many patients: this data teaches us what it is really like to be Cytosponged. Some points for consideration:  i. Clear background and detailed methods (including supplementary methods). Given the confusion outlined by participants of the association between GORD, Barrett's and Oesophageal Adenocarcinoma, could the first paragraph be reviewed to make crystal clear the associations? For example, I'm left with the questions, does Barrett's always precede Oesophageal Adeoncarcinoma and what else causes Barrett's? ii. Non-completers of the follow-up questionnaires differed from completers by age group, waist-hip ratio category and comorbidity status. The limitations sections points this out but doesn't comment on the potential implications of differences in these groups. Are there any? iii. Understanding test results data is relegated to Supplementary Table 2 and only given a sentence in the results. Could the authors consider giving these data more prominence? Especially, the sense of shock and the sense of confusion examples. How could the participant be protected from these feelings? iv. Satisfaction, in particular "Retrieving the Cytosponge". This section of the results (1.6.) seems relatively light on detail considering the satisfaction scores were some of the lowest. This is arguably the aspect of greatest concern to the lay reader/uninitiated. Could the authors offer more data to explore the reality of retrieving the Cytosponge? v. Anxiety. -pg10ln22- "efforts should be made to ensure patients know what to expect and are supported if they feel anxious" – could the authors elaborate on how to achieve this? Earlier in the paragraph it states "it will be important to provide high-quality information and manage patient expectations of the physical experience, as was done in BEST3" (ln15). It would be good to know more about how this was achieved in BEST3 without going to the protocol/report and whether this could be replicated in standard clinical (NHS) practice. The concluding section says something similar "will enable specific improvements to communications" – what are these improvements? vi. Perceived risk increased following Cytosponge testing and the authors imply this is to do with the delivery and quality of information received on the association between reflux, Barret's, and Oesophageal Adenocarcinoma. Would it be possible for the authors to elaborate on what they mean by "room for improvement" of the explanations given to patients". Is there anything we can learn from cancer screening or the risk communication literature that could move this statement to something more concrete? Thanks for inviting me to review this important piece of research.
--	--

VERSION 1 – AUTHOR RESPONSE

Reviewer: 1

Dr. Swarupananda Mukherjee, NSHM Knowledge Campus - Kolkata

Comments to the Author:

The stated objective of the work presented in the manuscript was Patient-reported experiences and views on the Cytosponge test: a mixed-methods analysis from the BEST3 trial.

While there was a large amount of data presented in the manuscript, there are SOME MINOR deficiencies and I have several questions and observations; as outlined below:

1. The part of discussion (INTERPRETATION PART) needs to be modified as the logic of this section is not clear and the contents are not closely related to the experimental results.

We thank you for taking the time to review our manuscript. Following your suggestion, we have amended the order of our arguments in the Interpretation section of the Discussion to follow a more logical flow and to reflect more closely how they were presented in the Results section.

2. It would be better if the author could provide more detailed information at the beginning (RATIONALITY OF THE OBJECTIVES) of the article about the previous approaches.

Thank you for this suggestion. We added some wording at the end of the Introduction to underline the difference between this study and the previous ones. This comment was dealt with jointly with comment 6.

3. Some spelling mistakes can't be overlooked. Few plagiarized words and sentences need to be modified.

We reviewed the manuscript and found two typos, which have now been corrected. Please note that we chose to write our paper in British English.

We have not found any plagiarized words or sentences apart from one sentence in the Introduction that was very similar to the one of another paper on the BEST3 trial recently published by our team (Swart N, Maroni R, Muldrew B, et al. Economic evaluation of Cytosponge®-trefoil factor 3 for Barrett esophagus: A cost-utility analysis of randomised controlled trial data. *Eclinicalmedicine*. 2021 Jul;37:100969. DOI: 10.1016/j.eclinm.2021.100969. PMID: 34195582; PMCID: PMC8225801). This was amended.

4. Figure No. 2 require better resolution AND caption need to be modified.

Figure 2 was submitted as an .eps (vectorial image), so the resolution issue in the pdf does not come from the figure itself but from its automated conversion by the online submission system. We thank you anyway for pointing that out. The caption is consistent with what is presented in the figure, but we have amended the wording to make it clearer.

5. (a) Risk compared to someone of the same age (comparative risk). (b) Percent absolute risk. from Fig 2 need to be explained clearly in the discussion part of the manuscript.

As mentioned above, the caption of Figure 2 has now been amended for increased clarity. We do not refer to Figure 2 in the Discussion section; we therefore added some further explanations of it in the Results instead (see section 3 Perceived risk of oesophageal cancer).

6. Novelty of the work should be discussed in the introduction part. Remaining part is ok and can be considered for publication.

Thank you for this suggestion. We added some wording at the end of the Introduction to underline the novelty of our work. This comment was dealt with jointly with comment 2.

Reviewer: 2

Dr. Brian Nicholson, University of Oxford

Comments to the Author:

Cytosponge offers significant promise to detect Barrett's Oesophagus in community settings remote from traditional invasive endoscopy. This mixed methods analysis examines in detail patients' experience of using Cytosponge, associated anxiety, and perceived cancer nested in the BEST3 trial, the largest trial of Cytosponge in primary care to date. These data are of vital importance as the thought of swallowing a capsule on a string then dragging a sponge back up the foodpipe is terrifying to many patients: this data teaches us what it is really like to be Cytosponged.

Some points for consideration:

- i. Clear background and detailed methods (including supplementary methods). Given the confusion outlined by participants of the association between GORD, Barrett's and Oesophageal Adenocarcinoma, could the first paragraph be reviewed to make crystal clear the associations? For example, I'm left with the questions, does Barrett's always precede Oesophageal Adeoncarcinoma and what else causes Barrett's?

Thank you - this is a very good point. Recent research shows that most oesophageal adenocarcinomas are indeed preceded by Barrett's. We have amended the first paragraph in the Introduction to make this clearer.

- ii. Non-completers of the follow-up questionnaires differed from completers by age group, waist-hip ratio category and comorbidity status. The limitations sections points this out but doesn't comment on the potential implications of differences in these groups. Are there any?

There may be indeed, so we have taken this suggestion and simulated a worst-case scenario in which these non-completers would have given the worst possible ratings to the IAPS and the STAI-6,

and used these conclusions to argue that the Cytosponge would still remain an acceptable test. These edits can be found in the Strengths and limitations section in the Discussion. Please note that the results of the STAI-6 at baseline for non-completers are already presented as a comparison in the Results section (2. Patient-reported anxiety before and after the Cytosponge test).

- iii. Understanding test results data is relegated to Supplementary Table 2 and only given a sentence in the results. Could the authors consider giving these data more prominence? Especially, the sense of shock and the sense of confusion examples. How could the participant be protected from these feelings?

Thank you for your interest in this data. We have included some more details in the Results section and specifically included some suggestions by participants on how to improve or change language in results letters.

- iv. Satisfaction, in particular “Retrieving the Cytosponge”. This section of the results (1.6.) seems relatively light on detail considering the satisfaction scores were some of the lowest. This is arguably the aspect of greatest concern to the lay reader/uninitiated. Could the authors offer more data to explore the reality of retrieving the Cytosponge?

We have now included a box describing the different types of discomfort experienced by participants during the retrieval step in the Supplementary materials.

- v. Anxiety. -pg10ln22- “efforts should be made to ensure patients know what to expect and are supported if they feel anxious” – could the authors elaborate on how to achieve this? Earlier in the paragraph it states “it will be important to provide high-quality information and manage patient expectations of the physical experience, as was done in BEST3” (ln15). It would be good to know more about how this was achieved in BEST3 without going to the protocol/report and whether this could be replicated in standard clinical (NHS) practice. The concluding section says something similar “will enable specific improvements to communications” – what are these improvements?

We added some more details on information provided to trial participants to manage expectations, how this could be achieved in standard practice and an example of improvements to communications to the Discussion.

- vi. Perceived risk increased following Cytosponge testing and the authors imply this is to do with the delivery and quality of information received on the association between reflux, Barrett's, and Oesophageal Adenocarcinoma. Would it be possible for the authors to elaborate on what they mean by “room for improvement” of the explanations given to patients”. Is there anything we can learn from cancer screening or the risk communication literature that could move this statement to something more concrete?

At the end of the Interpretation section in the Discussion, we now draw a comparison with patients' understanding of the disease in the cervical screening programme and we have added a reference to the risk communication literature.

Thanks for inviting me to review this important piece of research.

Thank you for taking the time to review it and for providing such insightful comments.